# Monitoring and Assessing Land Use/Cover Change and Ecosystem Service Value Using Multi-Resolution Remote Sensing Data at Urban Ecological Zone

Siqi Liu [1,2,3,*], Guanqi Huang [4], Yulu Wei [1,2] and Zhi Qu [3,5]

1    Institute of Land Engineering and Technology, Shaanxi Provincial Land Engineering Construction Group Co., Ltd., Xi'an 710075, China
2    Shaanxi Provincial Land Engineering Construction Group Co., Ltd., Xi'an 710075, China
3    Key Laboratory of Degraded and Unused Land Consolidation Engineering, the Ministry of Natural Resources, Xi'an 710075, China
4    Guangzhou Urban Planning & Design Survey Research Institute, Guangzhou 510060, China
5    School of Land Engineering, Chang'an University, Xi'an 710064, China
*    Correspondence: 4102090209@chd.edu.cn

**Abstract:** An urban ecological zone (UEZ) is an important part of a city, focusing on environmental conservation and ecological economic development simultaneously. During the past decade, the urban scale of Xi'an city in China has been expanding, and the population has been increasing rapidly. This dramatic change is a huge challenge to urban sustainability. It puts forward higher requirements for the construction of an UEZ. Under different spatial resolution scales, this study adopted Landsat8-OLI and gaofen-2 (GF-2) satellite high-resolution remote sensing data to interpret the land use/cover change (LUCC) of the Weihe River UEZ. The ecosystem service value (ESV) was assessed, and the ecological effect was analyzed based on LUCC. The results showed that the spatial distribution of land types in the Weihe River UEZ changed significantly from 2014 to 2020. The construction land gathered to the southeast. Especially, the vegetative land (i.e., forestland, grassland and other green land) and water body showed a slightly increasing trend since the official establishment of the UEZ in 2018. The cultivated land area gradually reduced, and the vegetative land area tended to be concentrated as well as expanded. Through the interpretation of GF-2 remote sensing data, the ESV at the Weihe River UEZ showed a downward trend in general. The high-value areas were mainly distributed in the Weihe River and its surrounding beach areas, which were greatly affected by river water scope. Construction land normally had low ESV, and it was affected by human activities obviously. Therefore, the development of urban construction had significant impacts on the Weihe River UEZ.

**Keywords:** land use/cover change (LUCC); ecosystem service value (ESV); multi-resolution remote sensing; landscape pattern; urban ecological zone (UEZ); urban construction

## 1. Introduction

Land is the basis of human production and living, as well as being the carrier of all kinds of resources [1]. Due to the interference from human activities, the land is facing increasing pressure [2]. According to the natural characteristics of land itself and its certain economic and social purposes, human adopts biological, physical, chemical and other technical means to launch long-term and periodic management on land [3]. Land use/cover change (LUCC) is an important way in which humans act on ecosystems. LUCC directly affects the landscape pattern, biodiversity and ecosystem [4–7]. With the acceleration of urbanization and industrialization, especially in developing countries, LUCC driven by human activities has led to the deterioration of the ecological environment and the degradation of ecosystem service function [8,9].

Different land use types have different effects on ecosystem services in terms of extent and pattern. Similarly, different land use intensity has different effects on the ecosystem [10]. Ecosystem services refer to all ecosystem products and services that contribute to human survival and living quality [11,12]. Ecosystem service value (ESV) is the benefits achieved from the ecosystem for human beings, directly and indirectly. It mainly includes the input of useful materials and energy to the economic and social system, the acceptance and transformation of waste from the system, and the direct provision of services to human society [13,14].

The research object of ESV can be a single landscape unit including forest [15], island [16], basin [17], and even an administrative scope, such as a specific city [18] or regions [19–21] of different scales. In 1997, Costanza [22] systematically assessed the functional value of global ecosystem services. Through quantitative evaluation of ESV, the variation trend of ecosystem structure and function had been revealed; moreover, the impact mode and degree of human activities on the ecosystem had been figured out [23]. The analysis of ESV is mainly realized by calculating the values of different ecosystem service functions in the whole region [24]. Subsequently, some researchers improved the valuation method of ecosystem services and proposed the equivalent factor method to calculate the ESV [25], as well as the derived static ESV [26].

In recent years, studies combining ESV with urban expansion and landscape pattern evolution have gradually emerged [27–29]. This research is of great significance for promoting land ecology and urban sustainable development. Landscape pattern, namely the spatial arrangement pattern of landscape elements, is an important approach to studying ecosystem quality and function. The temporal and spatial change of landscape pattern significantly affects regional biodiversity and other ecological indicators. It is an important method to analyze ecological quality change, which has direct impacts on ESV [30]. The main reason leading to landscape pattern change is the external interference, whose action mechanism is comprehensive, involving the interaction among human activities, nature and various organisms [31]. At present, the study on landscape pattern combining LUCC is commonly accepted. In addition, the selection of landscape index, its particle size and scale effects have also been a concern for a long time [32].

Many researchers focused on the prediction of ecosystem services and landscape pattern based on LUCC. Cellular Automata (CA)—Markov prediction model has been commonly used [33–35]. Meanwhile, evaluating landscape ecological risk and analyzing various ecosystem service functions based on LUCC, such as calculating habitat quality and carbon storage based on the integrated valuation of ecosystem services and trade-offs (InVEST) model [36,37], have become hot issues in this field.

A city is a complicated and integrated organization. The environment and development conditions of different urban regions inside the city vary wildly. It is insufficient to study only large-scale objects such as regions and cities. More importantly, it is necessary to study the small and medium-sized ecological areas in the city. However, such research studies are still relatively few.

The research on LUCC and ESV of urban ecological areas based on remote sensing is an important method to assess urban ecological environment quality scientifically. In order to ensure research efficiency and accuracy, remote sensing data with different resolutions should be adopted. Different types of multi-resolution remote sensing image data for research are complementary. Although the obtained remote sensing information has a certain redundancy, it is more accurate and comprehensive. Higher resolution images can make up for the shortcomings of lower resolution images in fine structure extraction, while lower resolution images have advantages in large-scale observation and rapid information extraction.

The different imaging principles of remote sensing images result in their various target characteristics [38]. Therefore, combined with Landsat8-OLI and high-resolution gaofen-2 (GF-2) satellite remote sensing data, this study aims at (1) analyzing LUCC and comparing the scale effect of the Weihe River UEZ in Xi'an on different spatial resolution



scales, (2) based on high-resolution remote sensing data, assessing the ESV and landscape pattern index, and analyzing the ecological effects, (3) putting forward some suggestions for the ecological development of Xi'an and the construction of the Weihe River UEZ.

## 2. Research Area and Data

### 2.1. Research Area

The Weihe River is the largest tributary of the Yellow River. It is one of the most important rivers in the Guanzhong region. It originates from Niaoshu Mountain, Weiyuan County, Gansu Province. It is 502.4 km long in Shaanxi Province, with a drainage area of 67,100 km$^2$, accounting for 32.6% of the total areas of Shaanxi Province [39].

The Weihe River flows through the loess area and carries great quantities of silt and sand. Most of the Weihe Basin belongs to the temperate continental monsoon climate zone, and the precipitation is concentrated in summer. The Weihe River mainly flows through the Weiyang district, Baqiao district, Gaoling district and Lintong district within Xi'an city [40]. The Weihe River UEZ is located in the north of Weiyang district, close to the south bank of the Weihe River, with a length of approximately 27 km and a maximum width of approximately 8.4 km. The total areas reach 120.78 km$^2$ (Figure 1).

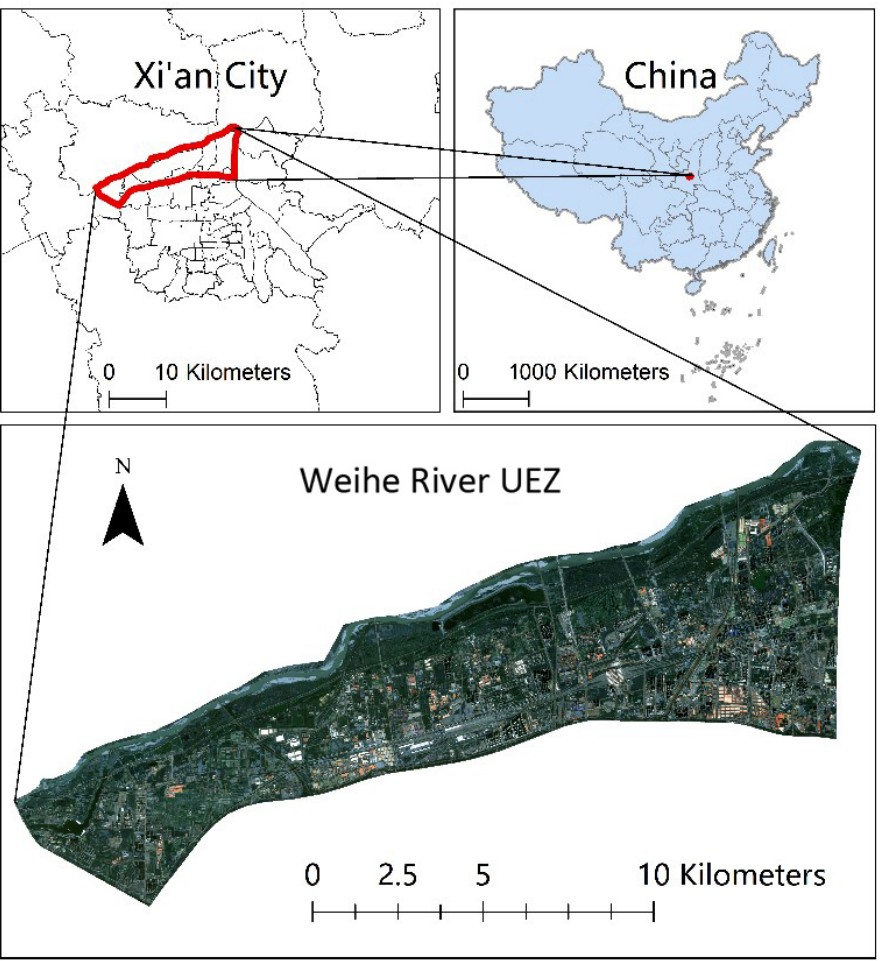

**Figure 1.** The location and satellite image of research area.

The Weihe River UEZ was built on the basis of a comprehensive treatment project for the Weihe River and was officially established by the government in 2018. In taking ecological protection as the main goal, this ecological zone is an important part of Xi'an sustainable development planning, as well as an important urban ecological barrier. It provides ecological services such as water conservation and eco-tourism for the city.

*2.2. Data Sources*

In this study, Landsat8-OLI data in 2014, 2016, 2018 and 2020 were obtained from the geospatial data cloud platform (http://www.gscloud.cn, accessed on 1 July 2022), one Landsat 8-OLI image data of the bottom cloud volume was used each year to cover the study area, and these data were all imaged between April and July. Landsat8-OLI can achieve global coverage every 16 days, with a spatial resolution of 30 m; origin data were preprocessed in Envi 5.3 (Radiometric Calibration and FLAASH Atmospheric Correction). The 2018 and 2020 GF-2 high-resolution data were obtained from Shaanxi Satellite Application Technology Center, one GF-2 image data was used in 2018, and two GF-2 image data were used in 2020 to cover the study area. Due to the difficulty of acquiring GF-2 data, the imaging times for the data in 2018 and 2020 were October and March, respectively, and the data used were low cloud volumes. GF-2 global revisit frequency not greater than 5 days, with a spatial resolution reaching 0.8 m (after preprocessing and fusion). The maximum likelihood classification was used to complete the interpretation. The kappa coefficient of Landsat8-OLI data interpretation was higher than 0.8, and the GF-2 data interpretation was higher than 0.85, which met the research requirements.

Due to the difference in spatial resolution between the two remote sensing data, the Landsat8-OLI data was divided into five categories, including unused land, water body, vegetative land, cultivated land and construction land, in this study. At the same time, the GF-2 data was divided into six categories, including unused land, water body, forestland, grassland, cultivated land and construction land. The grassland and forestland types in the GF-2 classification are uniformly covered by vegetative land types in the Landsat8-OLI data classification. The boundary data of the Weihe River UEZ was obtained from the planning map of Xi'an city and the vectorization of the planning scope of the Weihe River UEZ. Other vector boundary data were from the resource and environmental science and data center of the Chinese Academy of Sciences (https://www.resdc.cn, accessed on 4 July 2022).

## 3. Methodology

*3.1. Land Use Transfer Matrix*

The analysis of LUCC transfer is an important method to reveal regional human activities' impact on land use. Based on the land use data of the Weihe River UEZ interpreted by Landsat8-OLI and GF-2 remote sensing images, this study established the phase-by-phase transfer matrix in 2014, 2016, 2018 and 2020, respectively. Additionally, it compared the scale difference in the interpretation effect between Landsat8-OLI data with 30 m spatial resolution and GF-2 data with 0.8 m spatial resolution. The land use transfer matrix can scientifically reveal the overall land use structure and change characteristics, reflecting the change direction of land use types [41,42]. The formula is shown as follows:

$$S_{ij} = \begin{matrix} S_{11} & \cdots & S_{1n} \\ \vdots & \ddots & \vdots \\ S_{n1} & \cdots & S_{nn} \end{matrix} \tag{1}$$

where $S$ represents the areas of various land use types. $i$ and $j$ refer to the land use types at the beginning and end of the study period, respectively. $n$ is the total number of land use types. $S_{ij}$ is the area transferred from land use $i$ to land use $j$ during the research period.

*3.2. Landscape Pattern Index*

Landscape pattern reflects the spatial differences of patches in size, shape and attributes. Landscape pattern is usually reflected by various landscape pattern indexes. In this study, separation degree, fragmentation degree and dominance degree were selected to analyze the landscape pattern of the Weihe River UEZ. Fragmentation statistics (Fragstats) is widely used to calculate landscape metrics for categorical map patterns, of which Fragstats 4.2 is the most authoritative. Based on this platform, the landscape pattern index of the Weihe River UEZ was calculated pixel by pixel through the moving-window

method based on the GF-2 data in 2018 and 2020. Specifically, the selected window size was 100 m. The equations applied for calculation are shown as follows:

$$SPLIT = \frac{A^2}{\sum_{i=1}^{m} \sum_{j=1}^{n} a_{ij}^2} \tag{2}$$

$$F_i = \frac{np_{ij}}{ta_{ij}} \tag{3}$$

$$L_i = 1 - shei_{ij} \tag{4}$$

where Splitting index (*SPLIT*) is the splitting degree of landscape. $a_{ij}$ is the area (m$^2$) of patch $ij$. $A$ is the total landscape area. $F_i$ is landscape fragmentation, $np_{ij}$ is the number of patches of landscape type $i$, $ta_{ij}$ is the area of landscape type $i$. $F_i$ represents the fragmentation degree of landscape segmentation, reflecting the complexity of landscape spatial structure and the degree of human interference to the landscape to a certain extent. $L_i$ represents landscape dominance and $shei_{ij}$ is the Simpson evenness of landscape type $i$. $L_i$ is used to measure the deviation of landscape diversity from the maximum diversity, indicating the degree to which the landscape is controlled by several main landscape types.

*3.3. Ecosystem Service Value (ESV)*

ESV reflects the benefits that human beings can directly obtain from the ecosystem, which is closely related to the health and stability of the ecosystem and environmental policies. This study was based on the Arcgis 10.2 platform, using 300 m grid size, the ESV of the Weihe River UEZ was calculated in 2018 and 2020 using the GF-2 data by referring to the equivalent factor method and equivalent factor table proposed. Xie et al. considered that the equivalent ESV coefficient was 1/7 of the unit area value of market grain [25,42]. First, according to the grain yield, sowing area and market price (japonica rice, wheat and corn, obtained from Xi'an statistical yearbook and Shaanxi statistical yearbook) of prefecture-level cities in the study area, the unit area value of grain in the Weihe River UEZ was obtained. Second, the ESV of each category in Table 1 was calculated by the 1/7 multi-year from 2018 to 2020 average grain unit area value (equivalent ESV coefficient) multiplied by the equivalence factor. The data of equivalence factor referred to the studies of Xie et al. and Guo et al. [43,44]. Eventually, the ESV of the Weihe River UEZ in Xi'an can be calculated as Equation (5). The ESV calculations for both 2018 and 2020 are based on the same unit area ESV data (Table 1) since the same equivalence factor and multi-year average grain unit area values are used in the calculations.

$$ESV_i = \sum (A_i \times V_i) \tag{5}$$

where $ESV_i$ represents the ESV value of land use type $i$. $A_i$ is the area of land use type $i$. $V_i$ is the ESV per unit area of land use type $i$.

**Table 1.** Per unit ESV of various land use types at Weihe River UEZ (¥/hm$^2$/year).

| Primary Type | Secondary Type | Cultivated Land | Forestland | Grassland | Water Body | Construction Land | Unused Land |
|---|---|---|---|---|---|---|---|
| Supply Service | Food Production | 1539.08 | 507.90 | 661.80 | 815.71 | 0.00 | 30.78 |
| | Material Production | 600.24 | 4586.46 | 554.07 | 538.68 | 0.00 | 61.56 |
| Regulation Service | Gas Regulation | 1108.14 | 6648.83 | 2308.62 | 784.93 | 0.00 | 92.34 |
| | Climate Regulation | 1492.91 | 6264.06 | 2400.96 | 3170.50 | 0.00 | 200.08 |
| | Hydrological Regulation | 1185.09 | 6294.84 | 2339.40 | 28,888.53 | 0.00 | 107.74 |
| | Waste Disposal | 2139.32 | 2647.22 | 2031.59 | 22,855.34 | 0.00 | 400.16 |

| Primary Type | Secondary Type | Cultivated Land | Forestland | Grassland | Water Body | Construction Land | Unused Land |
|---|---|---|---|---|---|---|---|
| Support Service | Soil Conservation | 2262.45 | 6187.10 | 3447.54 | 631.02 | 0.00 | 26.16 |
| | Biodiversity | 30.78 | 6941.25 | 2878.08 | 5279.04 | 0.00 | 615.63 |
| Cultural Service | Aesthetic Landscape | 261.64 | 3201.29 | 1339.00 | 6833.52 | 0.00 | 369.38 |

*3.4. Spatial Analysis of ESV*

3.4.1. Hotspot Analysis

Based on ArcMap 10.2 platform, the hotspot analysis tool was used to test whether there existed significant spatial aggregation of high and low values of ESV in 2018 and 2020, using the GF-2 data in the Weihe River UEZ. The spatial distribution of ESV can be analyzed as a result.

$$G_i{}^* = \frac{\sum_{j=1}^{n} \omega_{i,j} x_j - \overline{X} \sum_{j=1}^{n} \omega_{i,j}}{S \sqrt{\left[ n \sum_{j=1}^{n} \omega_{i,j}^2 - \left( \sum_{j=1}^{n} \omega_{i,j} \right)^2 \right] / (n-1)}} \tag{6}$$

$$\overline{X} = \frac{1}{n} \sum_{j=1}^{n} x_j \tag{7}$$

$$S = \sqrt{\frac{\sum_{j=1}^{n} x_j^2}{n} - \left( \overline{X} \right)^2} \tag{8}$$

where $X_j$ is the attribute value of spatial unit $j$. $\omega_{ij}$ represents the spatial weight between the spatial units $i$ and $j$ (value is 1 when adjacent, 0 when not adjacent). $n$ is the number of spatial units. $\overline{X}$ is the mean value, and $S$ is the standard deviation. The statistical result of $G_i{}^*$ is Z points. Statistically significant positive Z-score indicates hot spots, and the higher the Z score, the closer the hot spots gather. At the same time, negative values indicate cold spots. The lower the Z score, the closer the cold spots gather.

3.4.2. Global Spatial Autocorrelation of ESV

Based on GeoDa platform, the spatial autocorrelation analysis of data is carried out to describe its spatial dependence and aggregation degree, and the global Moran's I index is selected for analysis in 2018 and 2020 using the GF-2 data. The specific formula is shown as follows:

$$I = \frac{n}{S_0} \frac{\sum_{i=1}^{n} \sum_{j=1}^{n} \omega_{i,j} z_i z_j}{\sum_{i=1}^{n} z_i^2} \tag{9}$$

where $z_i$ is the deviation between the attribute of element $i$ and its average value ($x_i - \overline{X}$), $\omega_{i,j}$ is the spatial weight between elements $i$ and $j$. $n$ is equal to the total number of elements, and $S_0$ is the aggregation of all spatial weights:

$$S_0 = \sum_{i=1}^{n} \sum_{j=1}^{n} \omega_{i,j} \tag{10}$$

$Z_I$-score is calculated in the following form:

$$Z_I = \frac{I - E[I]}{\sqrt{V[I]}} \tag{11}$$

where

$$E[I] = -1/(n-1) \tag{12}$$

$$V[I] = E[I^2] - E[I]^2 \tag{13}$$

### 3.4.3. Local Spatial Autocorrelation of ESV

The local indicators of spatial association (LISA) clustering map is used to evaluate the spatial clustering of land deformation in the study area in 2018 and 2020 based on the GF-2 data. When the LISA coefficient value is larger than 0, it indicates that there is a spatial positive correlation between local spatial units and adjacent spatial units, which is expressed as high or low. Otherwise, it indicates low-high or high-low, and there is a spatial negative correlation between local spatial units and adjacent spatial units. LISA was calculated by Equation (14).

$$I_i = \frac{(x_i - \overline{x})}{\frac{1}{n} \sum_{i=1}^{n} (x_i - \overline{x})^2} \sum_{i,j=1}^{n} \omega_{ij}(x_j - \overline{x}) \tag{14}$$

where $I_i$ is LISA of Moran's I, and *n* is the number of spatial units participating in the analysis. $\omega_{i,j}$ is the space weight matrix. LISA statistics of Moran's I index test is the same as the global Moran's I [45].

## 4. Results

### 4.1. Land Use Transfer and Scale Effect Analysis

This study constructed the land use transfer status of the Weihe River UEZ from different spatial resolution scales. Through the phase-by-phase land use distribution map and the construction of a land use transfer matrix, this paper analyzed the land use transfer from the spatial and temporal dimensions.

As shown in Figure 2, the construction land in the Weihe River UEZ was mainly distributed in the south and southeast. Additionally, with the construction of the ecological area, the aggregation degree in the southeast became higher. Green space was mainly distributed on both sides of the main stream of the Weihe River. Cultivated land was mainly distributed in a small part of the west, with concentrated spatial distribution characteristics.

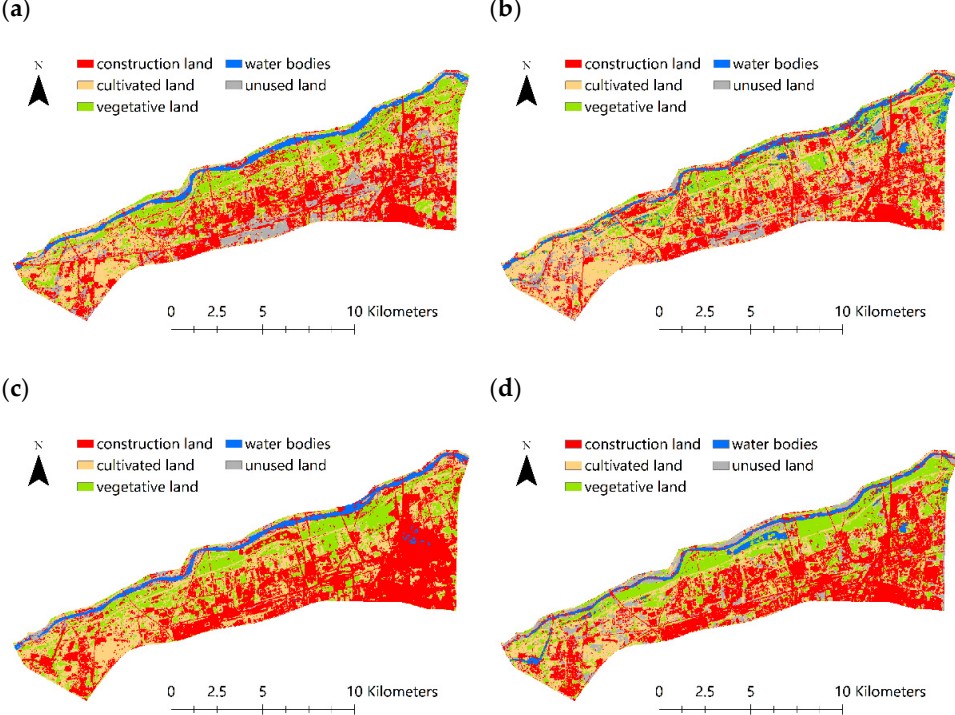

**Figure 2.** Landsat8-OLI land use classification map of Weihe River UEZ in 2014 (**a**), 2016 (**b**), 2018 (**c**), 2020 (**d**), respectively.

As shown in Figure 3, the construction land areas were all larger than 40 km$^2$ in four different years. This indicated that construction land was dominant in terms of land use types. In 2018 it reached its peak, 53.51 km$^2$, and then it decreased. By comparison, the areas of unused land and water body were relatively small in this UEZ. The water body areas were steady while the variation of unused land dramatically fluctuated during this period. In 2018, the areas of unused land reached the bottom, approximately 2.94 km$^2$. Moreover, it showed a remarkably increasing trend after 2018. Additionally, water body areas slightly decreased and cultivated land areas obviously decreased from 2016 to 2020.

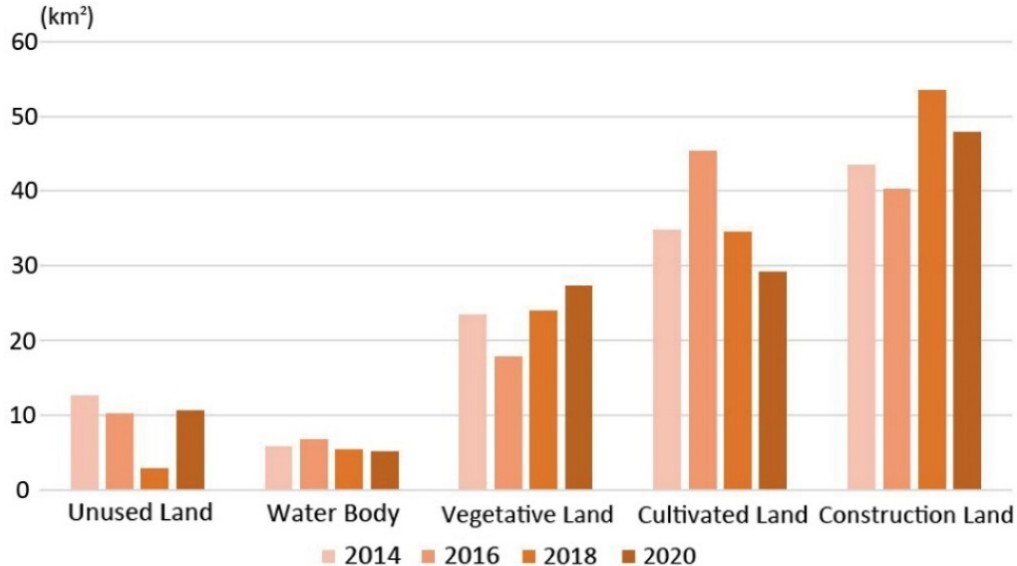

**Figure 3.** Landsat8-OLI interpretation of various land use changes from 2014 to 2020.

Table 2 indicates that the change of construction land converted to cultivated land was remarkable from 2014 to 2016, reaching 11.58 km$^2$. At the same time, the areas of vegetative land transferred into cultivated land were relatively large. This period was in the early stage of the development of the Weihe River UEZ. The construction land was still dominant in the research area, a part of which was converted into vegetative land. This variation implemented the official planning of the Weihe River UEZ. A large area of vegetative land transferred into cultivated land, indicating that the impact of locally cultivated land on ecology was significant from 2014 to 2016.

**Table 2.** Matrix of Landsat8-OLI interpretation of land use data from 2014 to 2016.

| From 2014 to 2016 | Construction Land | Cultivated Land | Vegetative Land | Water Body | Unused Land |
|---|---|---|---|---|---|
| Construction Land | 26.12 | 11.58 | 1.72 | 0.78 | 3.49 |
| Cultivated Land | 4.68 | 22.34 | 5.15 | 0.50 | 2.29 |
| Vegetative Land | 4.12 | 6.11 | 10.38 | 2.08 | 0.91 |
| Water Body | 1.61 | 0.33 | 0.37 | 3.38 | 0.15 |
| Unused Land | 3.85 | 5.18 | 0.26 | 0.03 | 3.40 |

According to Table 3, the transformation of cultivated land into construction land was obvious from 2016 to 2018. By comparing the land use map of the Weihe River UEZ, the cultivated land area was mainly concentrated in the southwest, and the transformation from cultivated land to construction land occurred in this region concurrently. In addition, the transformation from cultivated land to vegetative land cannot be ignored, and it reached 9.54 km$^2$. The increase of vegetative land reflected the development of the Weihe River UEZ and the improvement of local ecological quality from 2016 to 2018.

**Table 3.** Matrix of Landsat8-OLI interpretation of land use data from 2016 to 2018.

| From 2016 to 2018 | Construction Land | Cultivated Land | Vegetative Land | Water Body | Unused Land |
|---|---|---|---|---|---|
| Construction Land | 30.86 | 3.48 | 3.86 | 1.53 | 0.65 |
| Cultivated Land | 11.72 | 22.79 | 9.54 | 0.30 | 1.17 |
| Vegetative Land | 3.48 | 5.20 | 8.30 | 0.37 | 0.55 |
| Water Body | 1.68 | 0.33 | 1.32 | 3.13 | 0.30 |
| Unused Land | 5.90 | 2.80 | 1.13 | 0.14 | 0.28 |

According to Table 4, the significant change from 2018 to 2020 was the transformation from cultivated land to vegetative land, reaching 8.29 km$^2$. In addition, the area from cultivated land to construction land was also remarkable, reaching 5.91 km$^2$.

**Table 4.** Matrix of Landsat8-OLI interpretation of land use data from 2018 to 2020.

| From 2018 to 2020 | Construction Land | Cultivated Land | Vegetative Land | Water Body | Unused Land |
|---|---|---|---|---|---|
| Construction Land | 37.35 | 7.45 | 3.69 | 2.05 | 3.10 |
| Cultivated Land | 5.91 | 16.45 | 8.29 | 0.0045 | 3.94 |
| Vegetative Land | 3.12 | 4.64 | 14.59 | 0.07 | 1.71 |
| Water Body | 1.29 | 0.131 | 0.10 | 3.05 | 0.91 |
| Unused Land | 0.41 | 0.57 | 0.78 | 0.09 | 1.09 |

Landsat8-OLI data with a spatial resolution of 30 m was not accurate enough to interpret land use, especially in small-scale areas. It would be confused in the interpretation of features with similar pixel spectra, such as small and scattered water bodies and buildings. However, it was good in the interpretation of features with large continuous areas, such as vegetative land and construction land. Generally, it reflected the trend of construction land gathering to the southeast. The land use spatial distribution map (Figure 4) and transfer matrix (Table 5) of the Weihe River UEZ were further built based on GF-2 remote sensing images in 2018 and 2020, respectively, and the resolution and accuracy were improved as a result.

**(a)**          **(b)**

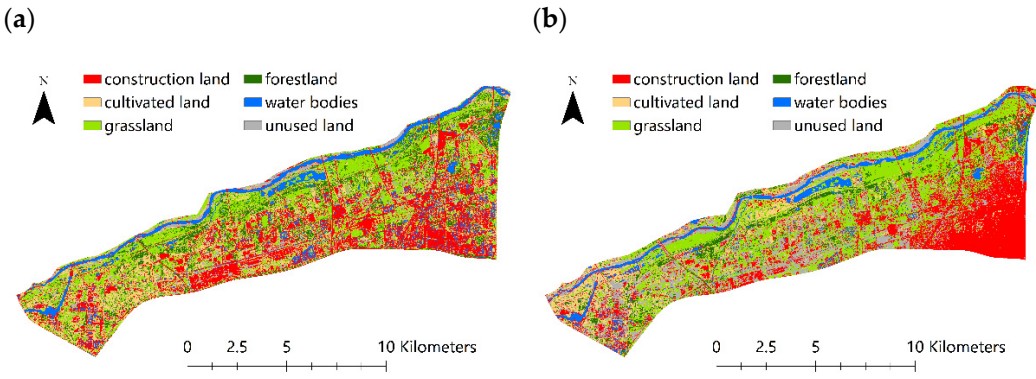

**Figure 4.** GF-2 land use classification results of Weihe River UEZ in 2018 (**a**), 2020 (**b**).

GF-2 land use classification results showed that the spatial distribution of construction land in 2018 and 2020 were concentrated in the southeast of the Weihe River UEZ. In terms of spatial agglomeration, the construction land in the research area is more obvious in 2020, while it was relatively scattered in 2018. Cultivated land was mainly concentrated in parts of western regions, and the largest land type was grassland.

**Table 5.** Matrix of GF-2 interpretation of land use data from 2018 to 2020.

| From 2018 to 2020 | Construction Land | Cultivated Land | Grass Land | Forest Land | Water Body | Unused Land |
|---|---|---|---|---|---|---|
| Construction Land | 12.67 | 2.89 | 7.45 | 1.55 | 1.15 | 5.86 |
| Cultivated Land | 6.37 | 3.73 | 7.03 | 2.03 | 1.59 | 4.83 |
| Grass Land | 6.26 | 3.96 | 9.88 | 2.20 | 2.94 | 4.98 |
| Forest Land | 2.85 | 1.93 | 4.74 | 1.34 | 1.31 | 2.12 |
| Water Body | 3.24 | 1.66 | 3.12 | 0.64 | 1.13 | 2.23 |
| Unused Land | 1.78 | 0.89 | 1.88 | 0.42 | 0.51 | 1.61 |

Compared with Landsat8-OLI, GF-2 interpretation data showed an obvious scale effect in the proportion of various land use types (Figure 5). It was difficult to deal with different classes in a single pixel when interpreting small-scale regional land use data using 30 m resolution remote sensing data. The classifier will select a representative object as the pixel land classification according to its mechanism, ignoring the fragmented and scattered land classification. The dominant, concentrated and continuous land classification will expand, eventually resulting in the error in the total area of each category.

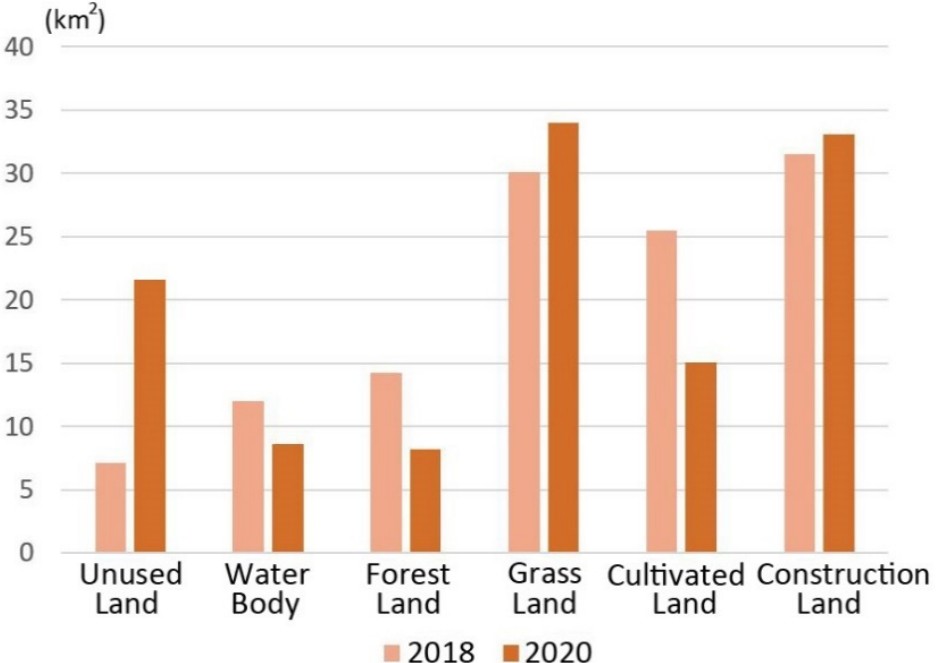

**Figure 5.** GF-2 interpretation of various land use in 2018 and 2020.

According to the interpretation of GF-2 data, the mutual conversion of land use types in the Weihe River UEZ was remarkable among construction land, grassland and cultivated land from 2018 to 2020 (Table 5). To be specific, there were 7.45 km$^2$ of construction land transferred into grassland, 7.03 km$^2$ of cultivated land transferred into grassland, 6.37 km$^2$ of cultivated land transferred into construction land and 6.26 km$^2$ of grassland transferred into construction land. The dramatic land use transformation reflected the instability of the current construction of the Weihe River UEZ. Meanwhile, it corresponded to the change of the spatial aggregation degree.

The transformation of land use in the Weihe River UEZ reflected the continuous strengthening of local ecological construction and the gradual improvement of ecological quality. However, due to the lack of interpretation of small area scale by Landsat8-OLI, the green space had not been classified in more detail, and the content covered by a single pixel and the information of similar spectral pixels were difficult to extract and classify. It was necessary to further utilize higher resolution GF-2 data for verification and analysis.

### 4.2. Landscape Pattern Analysis

According to the calculation of the landscape pattern index with the size of a 100 m window, the high-resolution indexes of spatial continuity of the Weihe River UEZ in 2018 and 2020 were obtained, as shown in Figure 6.

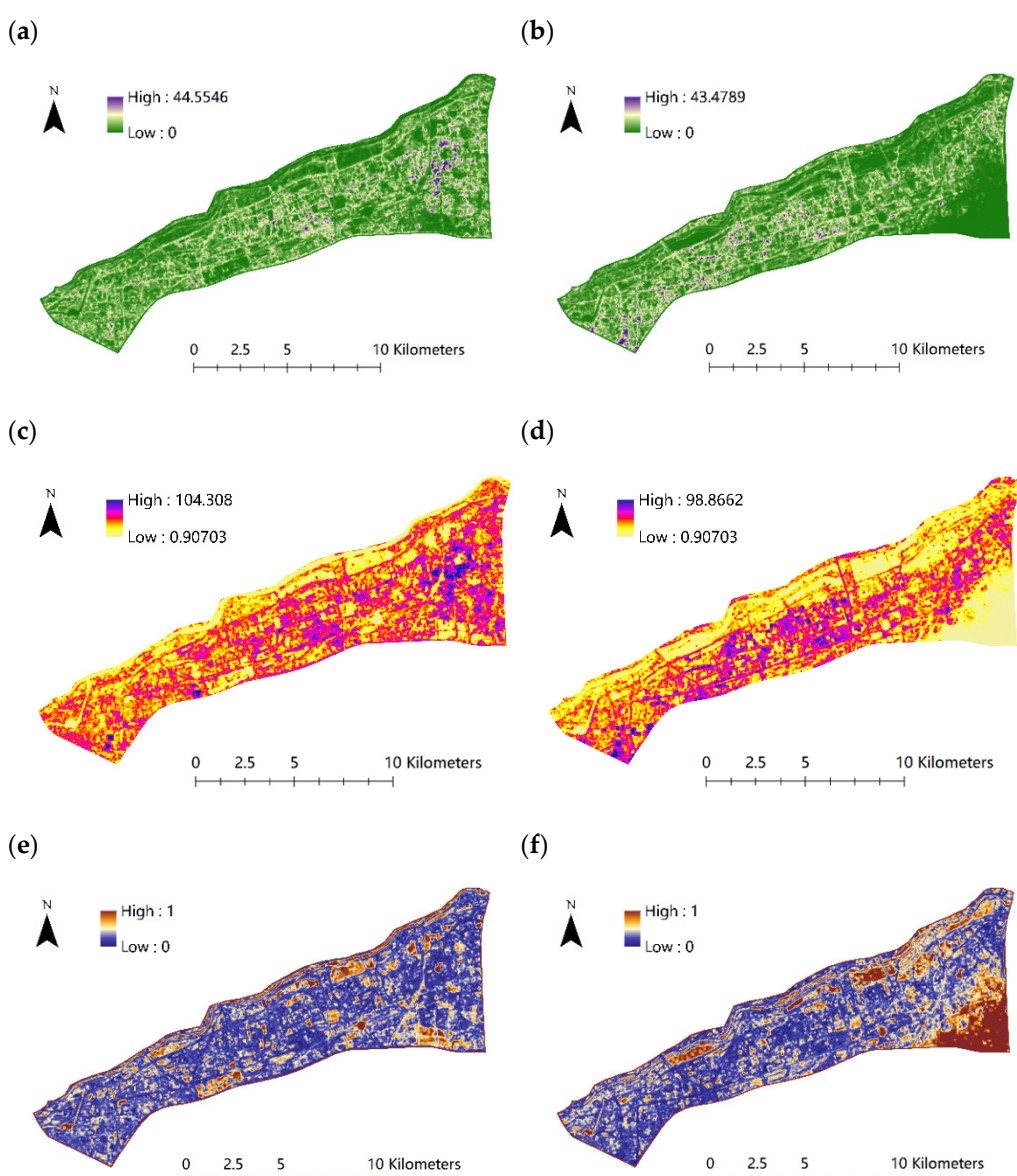

**Figure 6.** (**a**)The separation degree of Weihe River UEZ in 2018; (**b**) the separation degree of Weihe River UEZ in 2020; (**c**) the fragmentation degree of Weihe River UEZ in 2018; (**d**) the fragmentation degree of Weihe River UEZ in 2020; (**e**) the dominance of Weihe River UEZ in 2018; (**f**) the dominance of Weihe River UEZ in 2020. (GF-2 satellite based).

For different patches of a specific landscape pattern, the separation degree of individual distribution showed the spatial aggregation of landscape patches to a certain extent. From 2018 to 2020, areas with low separation degree concentrated in urban agglomeration areas, reflecting that the spatial agglomeration of construction land had a significant impact on the spatial distribution of separation degree. The regions with high separation had a trend of spatial dispersion from 2018 to 2020. Although the construction land showed a trend of spatial aggregation, the landscape separation degree of patches was relatively large within the construction land. It indicated that the spatial aggregation of construction land

still has a large potential for improvement and optimization. Fragmentation was an important indicator reflecting the degree of the ecological health of the landscape. From 2018 to 2020, the low-value areas increased significantly. The low-value areas were concentrated in the southeast of the Weihe River UEZ and in part of the river's south beach. Generally, the ecosystem in this area was relatively stable. From 2018 to 2020, the areas with high dominance increased, which were mainly distributed in the concentrated construction land. This result can be mutually verified with land use interpretation data. The construction land is dominant in the study area. Since construction land has a great impact on ESV and landscape pattern, the dominance of construction land needs to be controlled to a certain extent.

### 4.3. ESV Calculation Based on High-Resolution Images

ESV calculation results showed that the total value of ESV in the Weihe River UEZ was 227.46 million yuan in 2018 and 175.27 million yuan in 2020. It decreased by 52.19 million yuan. The average value of ESV in the Weihe River UEZ was 0.17 million yuan in 2018 and 0.13 million yuan in 2020. It decreased by 0.04 million yuan. According to Figure 7, the ESV value along the river was generally at a higher level and the highest level. The ESV in 2018 and 2020 showed an outward attenuation trend along the river. By comparing the land use data interpreted by GF-2 in 2018 and 2020, the ESV in the construction area was at the lowest level and lower level as a whole. The ESV with significant human interference showed a lower trend than that with less interference. It showed a trend of gathering to the southeast from 2018 to 2020 with the change of spatial distribution of construction and other land types. The phenomenon that high-grade ESV existing in the river areas showed was that the ESV generated by water conservation service was the main ecological service in the Weihe River UEZ. Compared with 2018, the ESV in 2020 showed a higher degree of spatial aggregation in low-value areas, which was conducive to environmental management and ecological governance.

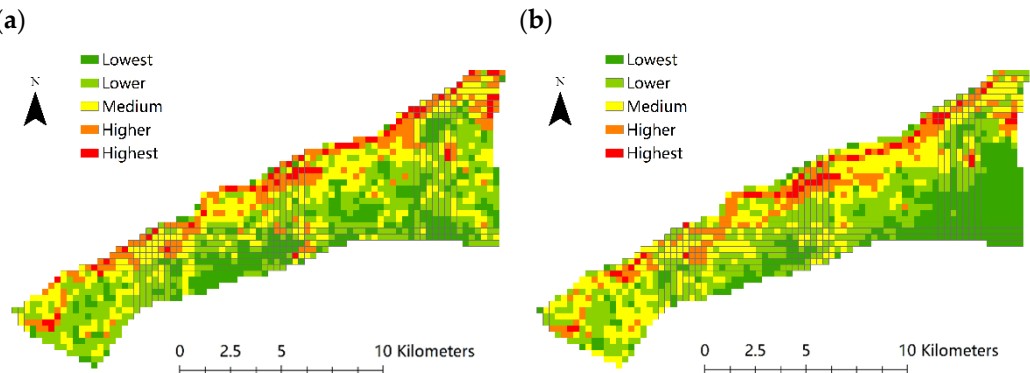

**Figure 7.** The ESV of Weihe River UEZ in 2018 (**a**) and 2020 (**b**). (GF-2 satellite based).

### 4.4. Spatial Analysis of ESV

In 2018 and 2020, the hot spots of ESV in the Weihe River UEZ were spatially distributed around the main stream of the Weihe River (Figure 8). Generally, the spatial relation of the cold–hot spot is stable. However, the variation trend in the cold spot area was relatively conspicuous. It was mainly determined by the change in construction density. The areas with insignificant cold and hot spots correspond to the grassland, forestland and cultivated land based on GF-2 interpretation data, in which the building density was low. It reflected that the distribution of construction land had an obvious density center. Especially in 2020, it showed a decreasing trend from southeast to outside.

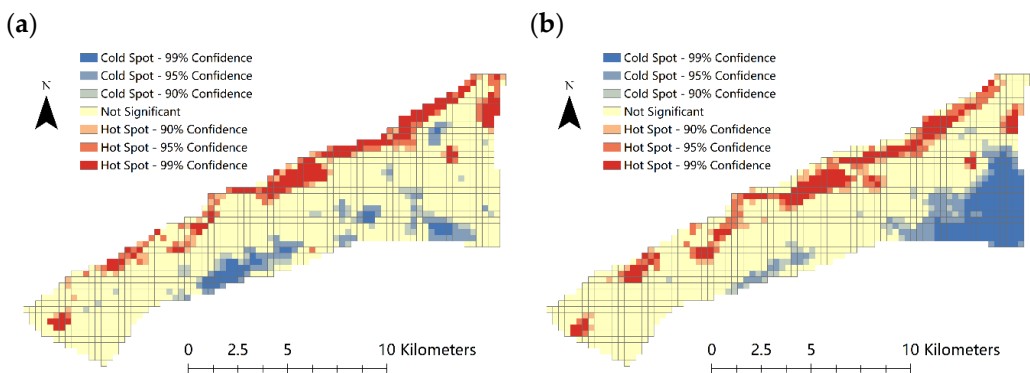

**Figure 8.** The ESV cold-hot spot of Weihe River UEZ in 2018 (**a**) and 2020 (**b**). The blue areas refer to cold spot in the 90%, 95% and 99% confidence interval. (GF-2 satellite based).

Based on the GeoDa platform, the global spatial autocorrelation calculation of ESV was carried out. The Moran's I index of ESV in 2018 and 2020 was 0.527 and 0.613, respectively, which was in an upward trend (Figure 9). Under 999 randomized displacements of the GeoDa platform, the *p* value was 0.001, which passed the significance test. Local spatial autocorrelation and its LISA map were used to analyze further the spatial autocorrelation of ESV (Figure 10).

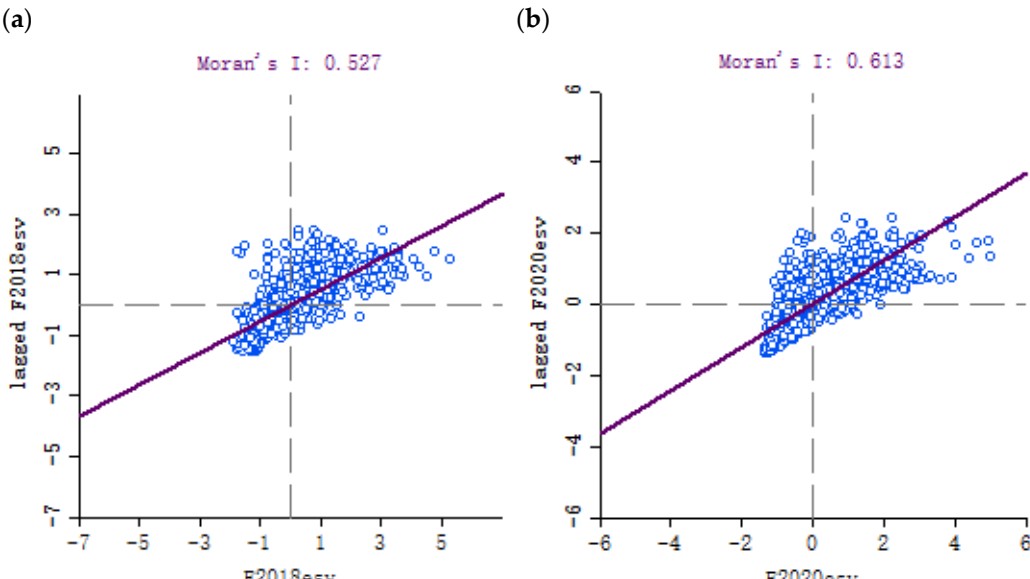

**Figure 9.** The Moran's I scatter diagram of global spatial autocorrelation of ESV at Weihe River UEZ in 2018 (**a**) and 2020 (**b**). (GF-2 satellite based).

According to Figure 10, the regions with obvious significance were similar to the hot spot analysis diagram, indicating that there was a significant spatial autocorrelation phenomenon in the cold–hot spot significant region of the Weihe River UEZ. The region occupied by insignificant areas included 796 grids in 2018 and 718 grids in 2020, which showed a decreasing trend. In 2018 and 2020, the positive spatial autocorrelation regions of high–high aggregation and low–low aggregation corresponded to cold and hot spots, respectively. Between the regional distribution of low–high aggregation and high–high aggregation along the river, there were few high–low aggregation regions and decreased continuously with time. It was mainly distributed in low–low aggregation areas.

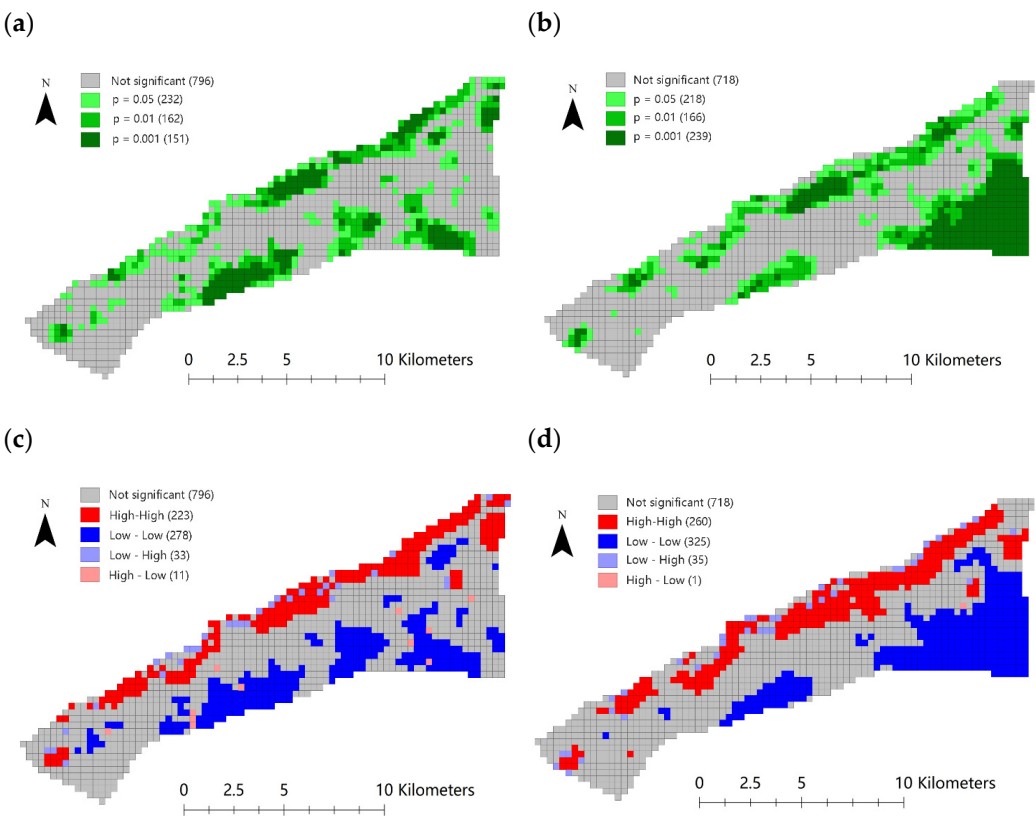

**Figure 10.** LISA significance map of local spatial autocorrelation of ESV in 2018 (**a**) and 2020 (**b**); local spatial autocorrelation LISA clustering map of ESV in 2018 (**c**) and 2020 (**d**). The blue areas refer to low–low aggregation type.

## 5. Discussion

### 5.1. Multi-Resolution Remote Sensing Method

The GF-2 high-resolution image can present more details, which can provide a reference for the selection of training samples when interpreting the Landsat8-OLI image in the same area. However, GF-2 is significantly affected by redundant information, such as building shadows during interpretation, causing a certain amount of blur and error in some information. It can be solved by post correction or adding training samples, but the efficiency is low. When using 30 m resolution remote sensing data to interpret small-scale regional land use data, it is difficult to deal with different classifications in a single pixel. The classifier will select a representative class as the pixel class according to its mechanism, resulting in the total area error of each class. The scale effect of remote sensing images with different spatial resolutions in a unified region has a great impact on the interpretation results. If the spatial resolution is particularly small, it will increase the difficulty and efficiency of training samples and interpretation, and the workload will increase exponentially. Therefore, it is suggested that the appropriate spatial resolution scale should be selected according to the research area and research targets. Moreover, the method and mechanism of fast processing high-resolution remote sensing images should be explored [46,47].

According to LUCC analysis by GF-2 remote sensing data interpretation, there existed a large-scale spatial aggregation phenomenon in the southeast of the Weihe River UEZ. Moreover, it transmitted to the ESV and its cold–hot spot results. However, this phenomenon did not appear in the landscape pattern index results calculated by using the moving window method. The reason probably is that the moving window has a certain fuzzy effect on the data. That was, the calculation process was not completely based on a single grid unit but took into account the values within a certain range around the central grid. Therefore, in the transitional boundary of the building aggregation area, the

calculation of its landscape pattern index was significantly affected by the grid values of non-building aggregation areas. If the size of the moving window were larger, the calculation results would be more blurred.

### 5.2. Planning Implementation and Policy Interpretation

According to the planning of the Weihe River UEZ, the government planned to build an education-housing cluster by establishing schools in the southeast areas. Such measures attracted more people to live in this region and drove economic development. From the point of view of land use, it was mainly reflected in the increase of construction land as well as the building density in the southeast areas. The concentration of urban settlement increases regional population density as well as human activity intensity. At the same time, the traffic efficiency and infrastructure utilization efficiency can be improved. Although the regional ESV decreases, it has positive impacts on reducing energy consumption and carbon emissions. This view corresponds to Song et al., who opined that high density, mixed land use and good connectivity of urban built environments are healthy and sustainable modes [48].

As shown in Figure 11, a large area of an ecological buffer zone, including protective green land and urban ecological parks, had been planned between the Weihe River and Xi'an downtown areas. Given the result of the research, scattered landscape patches gathered into a coherent large landscape belt, and the ESV in the north part had been improved from 2014 to 2020. Xu et al. pointed out that land resource was the main factor limiting urban sprawl. Urban construction had negative impacts on urban ESV [49]. Zhang et al. considered that urban parks could alleviate the impact of human activities on the ecological environment, especially in high-density cities [50]. The growth of urban parks is of great significance to urban sustainability. In the west part of the Weihe River UEZ, the continuous reduction of cultivated land indicated that urban expansion encroached on farmland. This phenomenon was spontaneous since these suburban areas are out of the planning scope. This spontaneous transformation of land use has potential threats, which should unify urban and rural planning, as well as strictly manage the non-agricultural conversion of farmland. In addition, the water of the Weihe River is obviously affected by local precipitation. The water flow varies greatly between different years and months. With many industrial factories relocated from downtown to suburbs, the water consumption, including agricultural, industrial and domestic water, use became quite large. This was consistent with the result that the water areas had decreased slightly. Therefore, the change in land use and ESV is just a symptom, and the key is to understand its causes and collective management behavior.

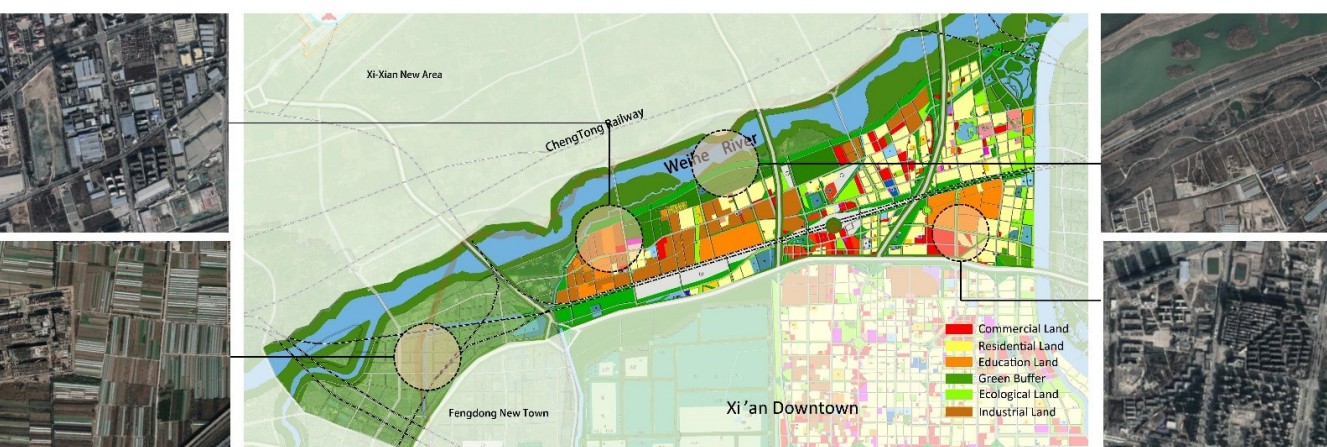

**Figure 11.** Land use planning and different land features including typical industrial area, farmland, ecological buffer zone and residential area. (Source: Google Earth).

## 6. Conclusions

From 2014 to 2020, the spatial differentiation of land types in the Weihe River UEZ was significant. The construction land gradually gathered in the southeast part of the research area. The bounds between construction land and other land types, such as forestland and grassland, had become clear. With the development of the Weihe River UEZ, the vegetative land, including forestland, grassland and water body, showed a rising trend as well as non-scattered features. The cultivated land area was gradually reduced. Additionally, the grassland area was obviously expanded based on GF-2 data interpretation. The construction land area expanded, and it was mainly concentrated in the southeast. In general, the ecological quality of non-construction land areas has been optimized. The zoning regulation made the boundary of different regions clear.

The total ESV decreased by approximately 23% from 2018 to 2020. In terms of spatial distribution, the ESV of the ecological area represented the high value was mainly distributed in the Weihe River surrounding beach areas. It was greatly affected by river water and riverside ecological landscape. The distribution of low-value areas was largely affected by construction land. From 2018 to 2020, the ESV high-value areas remained comparatively stable while the medium-value areas increased significantly. The landscape pattern showed that the landscape separation degree, fragmentation degree and dominance degree were greatly affected by the changes in the construction area, especially the landscape fragmentation degree. The construction land was greatly disturbed by human activities and the spatial structure was complex. The Weihe River UEZ in 2018 and 2020 had obvious global spatial autocorrelation. Moran's I was in an upward trend. The distribution of ESV cold–hot spot significant area was similar to local spatial autocorrelation significant area. The high–high aggregation of the hot spot area and LISA cluster was distributed between the main stream of the Weihe River, and the low–low aggregation of the cold spot area and LISA cluster was greatly affected by construction land. With the development of the land system in China, land use changes under the guidance of planning, staying a benign and sustainable state. Significantly, spontaneous land use and landscape change still exist, especially in urban–rural transition areas. This issue needs to be brought to the forefront of public discussion.

**Author Contributions:** Conceptualization, S.L.; methodology, S.L. and G.H.; software, Z.Q.; validation, S.L., G.H. and Y.W.; formal analysis, S.L.; investigation, Z.Q.; resources, Y.W.; data curation, G.H.; writing—original draft preparation, S.L.; writing—review and editing, G.H. and Y.W.; visualization, Z.Q.; supervision, G.H.; project administration, S.L.; funding acquisition, S.L. All authors have read and agreed to the published version of the manuscript.

**Funding:** This research was funded by Technology Innovation Center Funds for Land Engineering and Human Settlements, Shaanxi Land Engineering Construction Group Co., Ltd. and Xi'an Jiaotong University (No. 2021WHZ0090); Enterprise Innovation and Youth Talent Support Program of Shaanxi Association for Science and Technology (No. 2021-1-2); The Project of Shaanxi Provincial Land Engineering Construction Group (DJNY2022-29).

**Institutional Review Board Statement:** Not applicable.

**Informed Consent Statement:** Not applicable.

**Data Availability Statement:** The data presented in this study are available on request from the corresponding author.

**Conflicts of Interest:** The authors declare no conflict of interest.

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
