# Peer review of "Monitoring and Assessing Land Use/Cover Change and Ecosystem Service Value Using Multi-Resolution Remote Sensing Data at Urban Ecological Zone"

_sustainability, doi:10.3390/su141811187_

Round 1

Reviewer 1 Report

This is excellent research and worthy of publication.  It would benefit from clarification in language and definitions.  I am not up to date on the methods used, so do not understand as easily as contemporary peers in the field. As currently presented, it is very suitable for peer readership, less so for planners and policy types. Perhaps the latter needs another paper that focuses specifically on their needs and capacities.

Here are a few comments on presentation.  The beginning sections would be much clearer if broken into more frequent paragraphs. I saw logical breaks for paragraphs at lines, 44, 52, 69,74, 84, 92, 105 112 and 128, for example. I defer to your judgment on where the breaks should be, but they would really help clarifying the different ideas that currently are lumped too much in huge paragraphs.

I think every graph and table should be self-explanatory. Most of these currently are titled by method rather than substantive purpose.  Readers should be able to look at a graph or table and easily grasp what it is intended to show.  This can be improved.

The use of ESV needs clarification about its sources and methods.  This is more in my field. I did not get where and how the numbers were derived.  The references used are excellent. The paper would be helped , though, if there were some explicit explanation as to what was used and why.

One last point on ESV.  The central forces of concentration-diffusion have their own influence on ESV.  For example,  concentration of settlement reduces needs for human movement, for vehicles and fuel, with consequences for air quality and health.  Perhaps this doesn't matter in your case. If it does,  the overwhelming lessons of your work are in how distributions of human activity affect environmental qualities, yet I missed your attention to this relationship.

Overall, I really appreciated the quality of the work underlying the article.  I am also aware of my limitations with the methods.  Clarifications would be helpful and would strengthen the reach and durability of the article.  Some editorial attention might eliminate language elements that reduce accessibility.

I thank you for the opportunity to participate in the review, and recommend publication.

Jeff Romm

Reviewer 2 Report

Manuscript entitled Monitoring and Assessing Land Use/Cover Change and Ecosystem Service Value using Multi-Resolution Remote Sensing Data at Urban Ecological Zone” used high-resolution remote sensing data from Landsat 8-OLI and gaofen-2 (GF-2) satellites to interpret land use/cover change (LUCC) of the urban ecological zone (UEZ) of the Weihe River. Also, they evaluated the ecosystem service value (ESV) and analyzed the ecological effect based on land use/cover change. The authors present a solid work to evaluate the ecological effects of LUCC and ecosystem service value in urban ecological zone. Overall, the authors have collected massive data and performed a lot of analyses work. The discussion and conclusions of the study respond to and are consistent with the stated objectives; more than 85% of the citations are from the last 10 years. Therefore, the present work has an important implication for urban ecological zone protection, environmental conservation, and ecological economic development in cities. Please, find below my comments to the manuscript.

Major comment

(1) Section 2.2, Line 120-123, the authors do not clarify whether the data from both satellites (L8 and GF-2) were taken from the geospatial data cloud platform (http://www.gscloud.cn/). The authors do not mention which collection and processing level the Landsat 8 images were worked with.

(2) Section 2.2: The authors do not mention how many Landsat 8 and GF-2 images were used for each of the study years? Nor is it mentioned whether a specific season, date or month was selected for the study.

(3) The authors in the methodology in section 3.1 state the years and satellites used for Land Use Transfer Matrix. However, in sections 3.2, 3.3 and 3.4 they do not make clear whether they used GF-2 or Landsat 8 data, or a merge of both satellites for the years 2018 and 2020, since the results of these sections only show differences for these years.

(4) Table 5 (line 280): It would be clearer to the reader if in the matrix shown in Table 5 the classification used in the row is the same as in the columns (6 classification categories stated in section 2.2 for GF-2 satellite). The total sum of surface area of uses in the 2018-2020 study area from GF-2 does not match the total surface area calculated for Landsat 8 and stated in section 2.2 (line 111). The authors are recommended to review the land use and land use change data presented in the matrix in Table 5.

Minor comment

(1)   Section 2.2: The authors do not indicate the revisit frequency (temporal resolution) of any of the satellites over the study area. It would be interesting for readers to have this information.

(2)   Line 122: Fix the space between parenthesis and the parenthesis and coma “…cloud platform (http://www.gscloud.cn/), after…”

(3)   Line 206: Expand the acronym LISA when it is been used for the first time.

(4)    Line 244, Table 3: The table title indicates data from 2016 to 2018, however, in the first column the years are from 2014 to 2016.

(5)   It is recommended that the authors state in Figures 6, 7, 8 and 9 the satellite used to make the figures.

(6)   Line 304: Replace “there” for “the”: “For different patches of a specific landscape pattern, there separation degree…”
